# Expert Perspectives on the Effectiveness of Psychotherapy

**DOI:** 10.3390/ijerph20186739

**Published:** 2023-09-11

**Authors:** Henriette Löffler-Stastka, Andreas Ronge-Toloraya, Simeon Hassemer, Karl Krajic

**Affiliations:** 1Department of Psychoanalysis and Psychotherapy, Medical University Vienna, A-1090 Vienna, Austria; sdfhassemer@gmail.com; 2Postgraduate Unit, Medical University Vienna, A-1090 Vienna, Austria; andreas.ronge-toloraya@meduniwien.ac.at; 3Department of Sociology, University Vienna, A-1090 Vienna, Austria; karl.krajic@univie.ac.at

**Keywords:** Grounded Theory, theme analysis, expert interview, care effectiveness of psychotherapy

## Abstract

The effectiveness of psychotherapeutic care, as well as the implementation of adequate improvements, are in question. A qualitative interview study was carried out in a cyclical research design with a comparative analysis on the basis of thematic coding using Grounded Theory Methodology. An overview of the design, sampling procedure, and data analysis is given. A variety of critical perspectives emerged concerning the state of psychotherapeutic care in Austria. Two perspectives are presented in this paper as interim results: a health care administration perspective states a general lack of knowledge and a possible unmet need, problematizes the underutilized benefit of psychotherapists and describes a shift in regard to the issue of effectiveness of care to the topic of access to psychotherapeutic care and to a problem with the care and work ethics of professionals. In this perspective, one solution may be to implement intermediary organizations, clearinghouses with multi-professional teams, comprehensive documentation and an indication-oriented approach. The health insurance perspective also claims the organization-specific action problem and the lack of rules for clearing in such intermediary organizations, as well as the relevance of regulated, limited access to psychotherapy.

## 1. Introduction

A large observational study on the development of psychotherapy, psychotherapy trainees and training institutes was carried out by the Society of Psychotherapy Research in Austria (SPRISTAD study, [1,2,3]). The focus was on psychotherapeutic practice, on the way in which psychotherapy is provided and on the education of new psychotherapists and the development of their practical competencies, especially with regard to their role in the Austrian health insurance system. Concerning the practice of psychotherapy in the public health insurance system, inequalities in access to psychotherapeutic treatment were diagnosed. Social and regional inequalities and the availability of psychotherapeutic care, especially for patients with chronic mental illnesses, were problematized [2,4]. A “lack of effectiveness” of psychotherapeutic services [4], especially in serving the needs of vulnerable groups, was diagnosed [5,6]. Further, the role of organizations responsible for organizing psychotherapy care and professional training was criticized. Currently, health policy is initiating changes in legislation and the contracts offered to providers.

Overall, shortages in the supply of psychotherapeutic treatments and problems in the assignment of available resources have been discussed in Austria already over the last few decades, but with increasing dynamics in recent years. One part of this discussion refers to epidemiological changes, the prevalence of anxiety disorders, affective disorders or somatic symptom disorders increased in 2019, becoming the most frequent disease pattern in Austria [4], overtaking somatic diseases. While 23.8% of the Austrian population is considered to suffer from some form of mental illness, in epidemiological studies [6], only 14% of the population is treated in the health care system and only 3% receive psychotherapeutic treatment (1.5% psychotherapy and 1.5% psychotherapeutic medicine [3,4,5]). According to the OECD/EC and the Global Burden of Disease report, the follow-up costs of mental illness in Austria in 2019 were estimated to be 4.3% of the gross domestic product, i.e., EUR 13.9 Mrd. (OECD 2020 (2015); GBD 2019; Report of the Mental Disorders Collaborators, 2022 [4]). Due to the consequences of COVID-19, these costs are estimated to have increased up to 20% [4]. With regard to planned changes to the Austrian psychotherapy law and the upcoming redesign of the contracts with health insurance companies, a health policy controversy has unfolded. A widespread and prominent problem definition speaks of a ”lack of ‘care effectiveness‘“ of psychotherapeutic services in the outpatient sector.

However, it is often unclear what exactly is meant by the concept of “care effectiveness” by different health policy actors. Additionally, there are no systematic data on anything that could be understood as “effectiveness of psychotherapy” on health, especially population health. So far, this has not been measured in a broad sense in Austria, although there is a consequent plea for such an assessment [4,5,7].

This present study addresses this situation, that there is no clear concept for describing and evaluating psychotherapeutic care in Austria. This paper aims to display first the results of a comparative analysis of problem (solution) structuring as seen from different expert perspectives, which are linked to different organizational stakeholders in Austrian health care and health policy. 

### 1.1. Research Perspective and Topic Delimitation 

As a first step, our study aims to provide information on (1) the functions of psychotherapy and (2) the complexity of the psychotherapeutic care system in the Austrian context, which is an important background to understanding different options for defining “care effectiveness” more systematically.

Previous efforts to investigate the effectiveness of psychotherapy focused primarily on measuring care effectiveness by looking at the competencies of therapists [7]. Anzenberger et al. [8] defined “effective therapists” as those who are registered in the list of psychotherapists of the Ministry of Health (“BMASGK“) and were actively offering their services, understood as having treated at least one adult patient with a pathological disorder according to ICD-10 in private practice (compare also [7]).

Another important aspect mentioned in the discussion is the availability of psychotherapeutic care in the public health care system; i.e., as an in-kind benefit or at least a subsidized service. Both forms are considered to be relevant for accessibility for the less well-off. Another aspect refers to the types of cases treated by psychotherapists in private practice. 

From a curative, medical perspective, one would expect that psychotherapy should be primarily targeted at patients with a manifest disease. A previous publication, published by the Austrian Public Health Institute GÖG [7], reports—as a result of an online survey, in which nearly 25% of all active psychotherapists participated—that most patients (52%) receive a cost subsidy, about 27% receive in-kind psychotherapy and 21% are self-paying patients. So, at least those therapists participating in the survey seem to primarily contribute to the public health care system, although the share of in-kind therapies is only a quarter. 

This study also reports that 94% of the average total hours provided by the participating psychotherapists were reported to be spent on patients with a disorder defined as pathological [7]. This is interesting in the context of a discussion that criticizes psychotherapists as trying to avoid difficult cases and being not so much of a treatment but rather a health promotion measure, supporting subjective well-being or self-optimization of middle-class clients.

This lack of focus on the treatment of (psychiatric) disease is not a purely external critical perspective. As a matter of fact, it is argued by (parts of) psychotherapy itself that it contributes to improved health (in the context of health promotion).

It is also claimed that psychotherapy can and even should fulfill the prevention of mental and physical illness or the reduction of health risks as part of primary care structures and processes. 

In these two ways, psychotherapies appear as techniques for also changing behavioral parameters and/or characterological (mal)adaptive functions of those clients and their relationships, who are conceptualized in the health care system as symptom-free/non-sick persons; usually, they are suffering subjectively and report a burden, but not sufficiently strong or consistently enough to be considered clinically manifest symptoms.

Although there is a prevalent cultural assumption that psychotherapy is able to fulfill functions of health promotion and self-optimization, which—in a broader sense—should be considered a major contribution to public health, this present study accepts that currently a manifest disease or at least symptoms with “disease value” are the basis of public health insurance services. Given this, expectations of most health policy actors focus upon those functions of psychotherapy within the health care system that contribute to the treatment of/care for patients with manifest acute or chronic illnesses. Thus, we accept that the experts focus primarily on psychotherapy as a function of health care as well as in the context of secondary and tertiary prevention.

Following a social systems theory perspective on the health care system [9,10], we distinguish between three different system levels: First, a macro level that is relevant for defining legal, financial and organizational frameworks, which refers to function systems and includes lead organizations in Austria such as the ministry of health, health insurance funds, associations of therapy providers, etc. [10]. The second, the “meso level”, includes the organizational level of supply and demand, such as outpatient clinics, hospitals, and other service organizations that use psychotherapeutic interventions (constituting a wide range from employment services to prisons). Finally, we distinguish a micro level, which refers to the concrete interactions conducted by psychotherapists, psychiatrists, clinical psychologists, patients, case managers, etc. From this systems theory research perspective, we assume that the outlined system levels offer different problem (solution) possibilities or are limited to different problem (solution) possibilities. In this approach, psychotherapeutic care can be described in terms of how relationships and effects unfold from a range of (sub)systems, organizations and interactions with and within the health care system. Nevertheless, the system levels are interrelated and form relevant environments for each other.

Concerning the macro level, there is a multitude of different perspectives that can be considered relevant for psychotherapy. These perspectives can be located both in organizations that are usually understood as parts of the health care system but also in their relevant environments. These macro-level perspectives are relevant, e.g., for the design of framework conditions for the meso- and micro-levels in the form of laws (or reforms), central contracting systems, billing systems and policy programs for the redistribution of resources and also for systems of professional training. 

With these more complex environments, however, the concept of “care-effectiveness” becomes blurred and open to different interpretations. These interpretations are likely to be linked to interests dependent on the actor/organization and their specific functional area with reference to the health care system.

This current study takes this complexity and the assumed dependency of perspectives on care effectiveness on organizational interests as an opportunity and aims to focus on different problem descriptions and solutions. In sociological terms, the discussion about care effectiveness is consequently studied as a multi-perspective knowledge practice in which actors engage with reference to the macro-level.

### 1.2. Aim and Research Question

The primary objective of this research project is to conduct a comparative analysis of different professional and institutional perspectives on problem structuring and possible solution strategies. In particular, we include those perspectives involved in shaping the situation of psychotherapeutic care. We are interested in how experts from relevant areas process the problem description of a lack of psychotherapeutic care effectiveness.

Furthermore, we would like to know how the relevant experts evaluate the differentiation of psychotherapy care, which (missing) developments of rules and resources (structures) of psychotherapeutic care they problematize, which consequences they draw from their problem perception and finally, which (further) possibilities of development and problem-solving patterns of psychotherapeutic care they identify.

## 2. Design, Material and Method

A qualitative interview study was conducted in a cyclical research design. Here, we present the sampling strategy (compare also Figure 1), the survey method and lastly, the analysis strategy.

### 2.1. Sampling Strategy

The core research team (see Figure 1, core team: H.L.-S., S.H., K.K.) used a theoretical sampling strategy; according to which, the selection of cases is not determined ex ante, but is made in the course of empirical analysis and on theoretical understanding about the research object [11]. Accordingly, data collection was based on preliminary problem definitions, which were iteratively expanded from reviewing relevant publications, the literature and the context knowledge expertise of the members of the project team (core team).

Next, in a focus group (step 1), we supplemented the theoretical sampling with deliberate sampling by identifying strategically important individuals for the problem and in the field who had specific knowledge. The members of the focus group had a status in the field so that their implicit knowledge could be considered to display expert perspectives from the macro level. According to the cyclical design, we also went back to the international literature. For this first preliminary problem definition in the focus group (step 1), the challenge was to find out which actors are familiar with the institutional mechanisms of the field in question (the psychotherapeutic care system) or its relevant environments (e.g., training, financing, organization of services, legal regulation), and who could provide information accordingly. The focus group members considered to be experts (in step 2) are people who are respected as authorities on a topic and who also make use of this role. For this purpose, a search for organizations and actor perspectives along their function for psychotherapeutic care in the outpatient area (including strategic boundary perspectives of the inpatient area) was conducted (focus group step 2). Next, the core team formulated questions for the expert interviews.

We recruited 12 interviewees from various organizational backgrounds, which were interwoven with the health care system. The following list displays experts’ primary assignments, which are considered to be the main influence on their perspective.

Health Policy AdvicePsychotherapy Care and Professional RepresentationPatient AdvocacyPsychosocial Services and Professional RepresentationEconomic Advocacy and Social Insurance ExpertHealth Care Research and Health Policy AdviceHealth Care Research and Psychosocial Strategy DevelopmentHealth Care AdministrationPatient (Representation)Legal Affairs of Health InsurancePatient AdvocacyHealth Insurance Top-Management Perspective

### 2.2. Data Collection

Structured interviews with experts [12,13,14,15] were conducted in the presence of the interviewer or by videotelephone. Full anonymity was guaranteed to the interviewees when they were recruited according to the code of conduct for good scientific practice of the Medical University of Vienna. Informed consent was given by all twelve interviewees, and two declined the interview and one of them provided a written statement. Initially, 14 experts were asked to take part in the study. The data protection committee approved this study, and the ethics committee waived the consultation and approval of the study as no patients were involved. We were interested in operational knowledge as well as the contextual knowledge [13,14] of the experts. Operational knowledge refers to knowledge that claims validity for one’s own sphere of action. Contextual knowledge claims validity in relation to other fields of action in which the experts do not have the power to act [16]. 

In the interview protocol (see Appendix A), the subsequent questions were of interest: How is psychotherapeutic care described from the experts’ perspective, particularly in relation to the problematization of psychotherapeutic care and the problem description of a lack of care effectiveness? What are the interpretations and the explicit knowledge of the context (framework conditions, populations, developments, etc.) of the experts who are involved in the negotiation of decisions for the further development of the care situation? In the first part of the interview, we asked open-ended narrative prompts and thus gave the experts the opportunity to focus on their own topics and open up problem areas. Through follow-up questions, which were always adapted to the individual course of the conversation, the focus of the problem-centered interview [17] was then gradually adjusted towards answering our research question. Therefore, we also asked more confrontational questions. 

### 2.3. Data Analysis

The audio recordings of 12 expert interviews were transcribed. The transcripts were edited to safeguard the anonymity of the experts. The sample size was determined according to the saturation method. For a presentation of the interim results, we used the transcription technique of interview inventory according to Deppermann [18]. This step also provided an overview of the discursive process of the interviews. In addition, a protocol/data material was created that was suitable for further analysis through thematic coding or inventorying. 

The data were analyzed by the project’s core group (H.L.-S., S.H., K.K., A.R.-T.) according to the methodology and coding paradigm of Grounded Theory developed by Strauss [19]. An open coding approach generated initial concepts. Through recoding, categories understood as higher-order thematic concepts were identified. Transitioning to the form of axial coding, we generated further open concepts. In this process, the categories were placed in relation to each other and differentiation into categories and subcategories became increasingly possible. The technique of memo writing [20] accompanied the coding process in order to interrogate the categories continuously in their relevance to the research object. In order to reintegrate the expert perspectives as case determinations into the analysis, further analytical steps were taken following the model of thematic analysis, whereby categories were used as indicators for themes for the methodological integration of the two analysis perspectives. The central theme and analytical question was to what extent “within or between the (...) [expert perspectives] differences in themes or in the handling of themes” ([21], own translation) emerged. Thematic analysis helps to bring order to more manifest knowledge, as it is suitable to explore the “manifest content” as a “summing up presentation” or “a comparison of perspectives” [22].

## 3. Results: Perspectives on Utilization of the Concept of “Care Effectiveness” in Current Debates on Psychotherapy in Austria

Two perspectives on care effectiveness were selected for this paper: those of “Health Care Administration” and the “Health Insurance management Perspective”. These two were chosen due to their commonalities and differences. Within the Austrian health care system, the perspective of health care administration can be considered particularly important. The political system and, particularly, health politics are not only responsible for the legal framework but—at least in principle—also for essential parts of the financing and organization of health care. However, in this context, health care administration plays a moderating role rather than a governing role. The (public) health insurance management perspective has a more explicitly governing role—its responsibility for financing primary care also makes it strategically important for the governance of psychotherapy. Both perspectives are considered to be responsible for the current state of the public health care system, which makes a systematic analysis look worthwhile. Given the limited space available in this paper, we decided to start with these two perspectives on “care effectiveness” as a key concept of our research in depth rather than trying to provide an overview of all perspectives.

### 3.1. Perspective of Health Care Administration

From the perspective of the expert with a strong connection to health care administration, the question of whether psychotherapy is effective is considered to be relevant. However, this acceptance is also linked with the communication of a lack of systematic knowledge (“That is a good question. De facto, we don’t know too much about actual care effectiveness”). In this way, the question is implicitly considered to be outside of one’s organizational responsibility. So, it is also possible to accept a potential problem without coming under immediate pressure to act. Thus, a problem is articulated in which a relatively high prevalence of mental illness meets limited resources for its treatment. Therefore, the assumption of an unmet need is conceded. The description is “publicly funded treatment slots and estimated need [are] far apart.” The question of the need for treatment is linked to the offer of treatment options. The utilization of psychotherapy as a treatment seems justified insofar as there is a need for treatment. At this point, the need-based utilization is questioned. It is assumed that psychotherapy is used even though it is not goal-oriented or not suitable for solving the problem. It is claimed that existing resources are used up, which are then no longer available for cases worthy of treatment “Psychotherapy is not the solution for all PSY problems”.

The global argumentation pattern of an “unmet need” establishes the possibility of seeing the goal of the thematization of care effectiveness in problematizing under-exploited capacities of psychotherapists, clinical psychologists and health psychologists.

The problem definition of an unmet need shifts in the course of argumentation to a complaint about the care and work ethics of the professionals (“If all those who have the professional license and those who are registered in the professional register would really do this job, then we would even have an overprovision in Austria”) and, further, to questioning the access to care as “the way in which services are made available”. 

According to the expert, a shift of responsibilities is recognized: the access problems to care are considered a consequence of the quota system for fully financed psychotherapy treatment. The quota system entails the availability of only a pre-defined number of publicly financed therapy slots leading to the situation that patients who have a medically indicated need for therapy do not receive it because therapy slots are already occupied. As a solution, the implementation of clearinghouses is suggested in order to sort out the unnecessary accesses and to distribute the needs in the system in a more targeted way. This can be seen as a possible solution to the problem and as a new problem for action. However, the significance of this view is corrected by two descriptions of the problem, about which there is a lack of knowledge: on the one hand, the relevance of the question is emphasized in the perspective that “we know too little, whether it is the more skilled or those who depend on it… who receive treatment”. There is such an unconfirmed but propositional knowledge of the fact that the chances for psychotherapy places might be unequal with regard to target group milieus. On the other hand, the focus on equal access to care is expanded with the issue of effectiveness and quality assurance of medical treatment (“Another question here is whether the therapeutic services actually lead to the desired improvements”). 

Within this perspective, the issue of the effectiveness of care thus becomes a description of a complex problem constellation. One possibility of solving the problem in the sense of disambiguation is seen in developing an evaluation system. A proposal obligates therapists to fulfill “certain documentation requirements”, the “results of which flow into a register”, in order to “know which treatments actually have an effect on which people, for which indications”. An advantage of this evaluation that can be specified for target group and indications is seen not only in generating knowledge about the effectiveness and efficiency of treatment (especially treatment time, outcome) but also in patient safety. A necessity is defined to identify “black sheep among the therapists”, and there is also an expectation to build regional knowledge for clearinghouses (“which help would be particularly useful for this person with this indication in this region now”).

At this point, the topic of clearinghouses is again linked by the sub-topic of multiprofessionality. This is because the registry is also associated with the task of gaining expertise on “the combination of which professions would need to work together,” which should be helpful to ensure the quality of collaboration between health professions. A reference to the debate about primary care centers (currently in development in Austria) is also anchored within this context. Relevant topics include an expectation for low threshold access to psychotherapy, especially in association with the spatial concentration of different medical services. Reference is made to a reduction of stigma (“the inhibition threshold decreases to start a therapy, if I can also do it in the same building (…). But to go to the primary care center where there is easily a room where psychotherapy is also offered also helps some people who are afraid of stigmatization”). The problem-solving issue of effectiveness and quality assurance is closed by the issue of financing (“and yeh, but it’s all very expensive” and the cancellation of specificity (“in Austria unsatisfactory also for somatic diseases”). 

If one includes the broader context of the conversations, it becomes clear that for health administrators, this goal of more comprehensive documentation is linked to an action problem of their role in health planning. The problem lies in the dependence on research (“We can only rely on research to tell us there are certain evidence-based reasons”), for example, to have an expertise vis-à-vis public financiers and not to be limited exclusively to the bringing together of players and their moderation; “the Ministry of Health has only a moderation function yes it can bring the players together (...) it has as a ministry no direct role in organizing or controlling effective psychotherapeutic care yes we can not say [that] we [will] now implement something ourselves”.

### 3.2. Health Insurance Perspective 

Another perspective relevant for the design of psychotherapy care and its effectiveness is offered by the top management of public health insurance. Its main function is financing primary and specialized care outside the hospital sector and providing low threshold access through contracts with a limited number of providers. In psychotherapeutic care, health insurance offers a very limited amount of free therapy places and also co-payment for therapies in cases with an explicit psychiatric diagnosis and an explicit prescription.

In the perspective of the expert that was interviewed, the issue of effectiveness of care is directly linked to the relevance of regulating access to psychotherapy (“it is important to me to regulate access to psychotherapy”). In the context of the topic of access regulation, a supply problem for specific target groups is assumed, according to which the “wrong people utilize psychotherapy”. One of the problem descriptions offered is that several patients are medically under-diagnosed and under-treated with psycho-pharmaceutics. The assumption is that effectiveness of psychotherapy often depends on its combination with psycho-pharmaceutics, which is considered “sometimes simply necessary to make psychotherapy successful.”

In this perspective, this is used to argue against granting free and direct access to psychotherapy. In this understanding, the current system is too liberal: “Currently everyone can go to a psychotherapist, can choose one, can say I want psychotherapy”, and this creates financial pressures for health insurance: “of course the patients will then apply to the health insurance for cost coverage”.

What would seem more rational is that “access were not designed to be so free”. In this perspective, free access is reinterpreted as randomness and contrasted with a proposed solution to establish a new structure named “clearinghouses” (“we don’t have this type of clearinghouse in Austria yet, but randomness”). This is because people are assigned psychotherapeutic care based upon which kind of psychotherapy they are “willing, considering suitable, or feeling capable” of receiving, which presupposes competencies of rational selection based on judgment to distinguish illness value, load and social problems. The assumption is that such an indication-oriented view cannot be expected from the individual patient but would have to be organized for the planning of psychotherapy (care), in general.

For the health insurer, however, care effectiveness is not only characterized as efficiency in terms of treatment duration. Care effectiveness is also seen as the effectiveness of a treatment for a specific clinical condition and as the effectiveness for all patients with that condition. Scientifically proven effectiveness is also argued as a sine qua non condition for payment of treatment by health insurance “the legislator also says, any treatment [that] we fund as a public institution must be scientifically evidence-based”. 

This refers to a long-standing and deep skepticism of Austrian public health insurance concerning the effectiveness of psychotherapeutic treatment and the equivalence of different psychotherapeutic methods. The lack of consensus among expert voices concerning the effectiveness of psychotherapy is contrasted in this perspective with the assertion that treatment with antibiotics always works for every patient in the same way, and is also independent of the personal characteristics of the patient and doctor. As another example of a much more “objective” practice in medical care, a good personal relationship with a radiologist or even a family doctor is considered to be much less relevant for therapeutic success than in psychotherapy.

This deep skepticism towards the undoubtedly larger role of the patients’ subjectivity in psychotherapy is expressed in the way the access problem to psychotherapy is phrased—the central issue is not to assure a sufficient supply of free or at least co-financed psychotherapy slots to match demand, but “to bring the right patient to the right psychotherapist”. From a public health insurance perspective, “care effectiveness” seems primarily threatened by a lack of rationality in the utilization of available resources. In this perspective, the diagnosis of a “lack in care effectiveness” relates primarily to problems of misuse (or even overuse) rather than underuse of (publicly financed) psychotherapy. As a solution, explicit and sharp gatekeeping is considered to be needed for psychotherapy, e.g., in the form of “clearinghouses”, which should be able to distinguish between illness valence, individual psychological burden and social problems, and assign the problems to the most qualified carer. This is considered a technical problem of “clustering apart” clients/patients on the one hand, according to the nature of their problems, and providers on the other hand, according to their proven effectiveness in dealing with specific problems. In this perspective, the internal heterogeneity of psychotherapeutic methods asks for organizing them into psychotherapy methods clusters to counter professional interests and to avoid transparency (“there are many schools out there”). 

In the context of this argumentation pattern, a reformulation of the health care problem is attempted in which low-threshold access to psychotherapy is perceived to be a problem rather than a solution. Only in metropolitan areas, such as Vienna, is a (relative) lack of supply reported, although the number of potential providers is much larger than in rural areas. The number of people that make use of psychotherapy is much larger in urban areas, but this leads to sub-optimal results from the health insurance perspective. “People with minor problems and illness can access psychotherapy more easily than people with more severe problems of real mental illness“, which should be the primary target group for publicly funded psychotherapy.

This “missing out the target group” diagnosis is strengthened in this perspective by the work habits of professionals: many psychotherapists “just look for the less severely ill patients because they want to have less burden, or have to manage their burden, and the more difficult patients often fall by the wayside”. This assumption of a work ethic of the members of the profession that is considered problematic is used to question the behavior of the professional groups in political discussions (“there is always shouting we need more psychotherapy”). In addition, professional representatives do not seem to know what the patient really needs (“there is too little differentiation between individual forms of treatment, which the patient really needs”).

The problem attribution on a micro level (problematic preferences in treatment decisions by patients and therapists) and a macro level (lacking differentiation and wrong conclusions in political discussion) is put into the problem-solving context of the “clearinghouses” mentioned above and should solve problems of misallocation by limiting patients’ choices to two or three psychotherapy offers and oblige treatment providers to accept the patients assigned to them.

The positioning of psychotherapy in the health policy discourse is further weakened—as seen from the health insurance perspective—by perceived deep dissent between various groups offering psycho–social services concerning needs and also competencies of different providers. This refers not only to different schools of psychotherapy but also to clinical psychologists and health psychologists, as well as psychiatrists. From a health insurance perspective, skepticism seems especially strong towards psychotherapy, whereas health psychologists (as well as social work) are attributed to have the potential to prevent mental illnesses (for example, in a target group of children and adolescents).

The health insurer’s perspective on care effectiveness can ultimately be tapped against the background of the consequentialist processing of an environmental problem that threatens the capacity to finance services. What is striking here is that the issue of the need for psychotherapy is not litigated in the immediate context of the issue of care effectiveness. In the same way, the organization-specific action problem (“as health insurance we want to establish uniform care”) of harmonizing health insurance is not articulated in the framework topic, care effectiveness. In this respect, questioning care effectiveness becomes a strategy of not naming thresholds of eligibility for health insurance-financed psychotherapy.

## 4. Discussion 

This paper provides information on design, methods and the first results of an expert interview study to answer the question of how the situation of psychotherapeutic care is currently processed in health policy discussions. For this purpose, 12 experts from different relevant macro-level fields were interviewed. Over the course of these interviews, the current situation of psychotherapy care in Austria was addressed from different professional and institutional perspectives.

In current debates, a diagnosis of a questionable “care effectiveness” plays an important role. Therefore, this has been selected as the focus of this first paper on results. We described two perspectives on psychotherapy care effectiveness central to regulating and public financing: health care administration and public health insurance. 

The results thus provide indications that, depending on the perspective, the diagnosis of a “lack of care effectiveness of psychotherapy” makes it possible to process different health care problems, environmental problems and problems of action at the macro level. The analysis indicates, for example, that even between these two experts representing actors with a clear health policy agenda, a common understanding of the topic of a lack of care effectiveness is difficult to identify. 

From the two perspectives analyzed in this paper, a working hypothesis emerges that is relevant in the continuation of the analysis, which will be further tested. On the basis of the topic of the effectiveness of care, the case portraits primarily process the (lack of) ability of the respective actors to act in shaping the framework conditions of psychotherapy care by the experts’ organizational background. 

## 5. Limitations and Outlook

A major limitation was the limited number of expert perspectives we were able to include in the research project. When performing the steps in our theoretical sampling, it quickly became clear that we would not be able to cover all the potentially relevant differences of perspectives in our study. This was partly due to the very limited funds we had for carrying out the fieldwork, but also to a reluctance of many experts to co-operate with the researchers. Given the long history of implementation of publicly organized/financed psychotherapeutic care in Austria with embittered conflicts and controversial problem descriptions, we were not too surprised of this reluctance, but of course, the empirical basis for each of the perspectives we tried to include is limited and asks for further research to analyze.

The difficulty of reaching experts can provide initial indications of their social status and the attempted relevance of their perspective to control their organization and institutional knowledge. For this purpose, some potential interviewees already resorted to a repertoire of defense techniques when making contact, which can be well described following Lau and Wolff [23]:

Some informants speculated in this context on what we interpret as task completion by waiting. In the case of a number of social insurance officials, for example, we were able to register how the request for contact was left unanswered, which can presumably be interpreted as a strategy of completing tasks by waiting. Others, however, tried to avoid having to take action themselves by assigning alternative interview partners while at the same time accepting the topic and the question. In the area of health administration, but also patient representation [24], we registered that the research request was accepted, but by assigning alternative interview partners and splitting the research interest, a strategy was possibly pursued to produce a more advantageous presentation for the organization.

## 6. Conclusions

With the results, a basis is created, of which a further and deeper investigation on the topic, care effectiveness of psychotherapy, is possible. In this context, it is planned to focus on the presentation of further detailed accounts of the main perspectives that emerged.

As is so often the case, the thematic commonality of all the informants surveyed can be seen in the fact that the question is how the complex environmental experience can be transformed from not knowing into knowledge and shaping the framework conditions of psychotherapy care. 

It is not just about the fact that data are missing, but also that there is no agreement on what an adequate survey instrument should look like with regard to the concept of care effectiveness. This theme of the lack of knowledge also extends to the dimension of content when the lack of appropriate documentation practice within psychotherapy practice is repeatedly identified [25,26,27], which, in turn, is associated with the hope of obtaining a more resilient data and knowledge base in order to then, in turn, follow state regulation, which is the necessary transparency for further public funding and the associated requirements for continuous evaluation of therapy efforts.

The knowledge theme also comes in another form as a unifying element with regard to the educational system perceived in this perspective as the environment of the care system. Here, too, the issue of the lack of knowledge is opened up as a quality feature or deficiency in the very different training opportunities with varying admission requirements available to date. Especially, if psychotherapy is to be taken seriously as a treatment for the sick, as distinct from life counseling and “first world problems”, the requirements for training and licensing regulations must be harmonized at a high level [28]. However, the experts in question also point to the difficulty of exerting (controlling) influence on environmental problems.

For further research in the area of professional interest politics, we were able to register techniques of inquiring and delimiting, especially in the case of a professional representation, with the former technique, for example, asking the researcher to reformulate the research goal and their approach, and the latter technique enabling an internal organizational review. Implicitly, a shift from expert knowledge to functionary knowledge happened, whereby the interview request can be rejected as thematically inadequate for the organizational goals.

The fact that such strategies were used should be understood as part of the analysis and not be regarded as normative or judgmental. Especially, in sensitive areas in connection with major changes that are in process, a diplomatic approach on the experts’ side is not surprising. We are very grateful to all interviewees for taking the time and effort to answer our questions. 

The interim results remain at a level of empirical generalization [29], in that we make statements about the structuring of expert knowledge based on the different perspectives. However, the exemplary character does not only concern the analysis but also the need for further surveys (e.g., if free access to psychotherapy or filtering according to a specific indication process provides care effectiveness, etc.). Nevertheless, we think that a more detailed presentation of the perspectives can give differentiated insights into expert discourse regarding the care effectiveness of psychotherapy.

## Figures and Tables

**Figure 1 ijerph-20-06739-f001:**
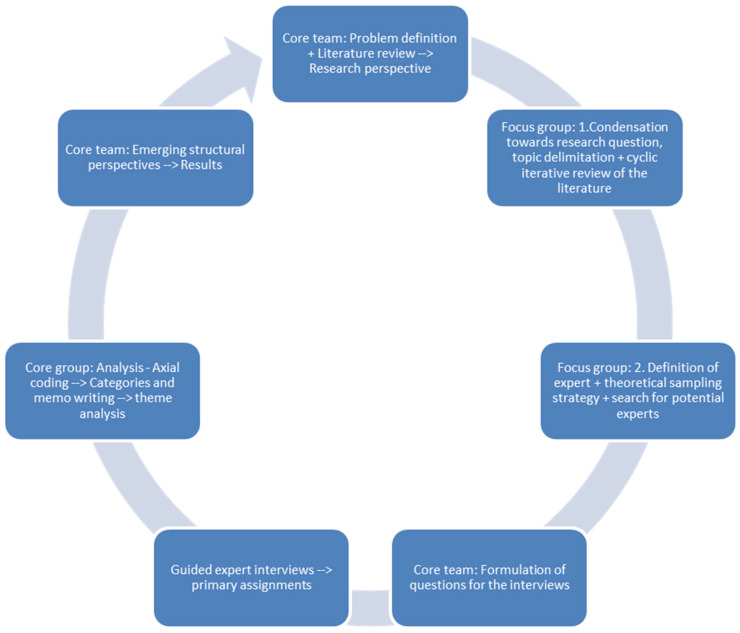
Study process.

## Data Availability

Data is unavailable due to privacy or ethical restrictions.

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
