# Peer review of "Expert Perspectives on the Effectiveness of Psychotherapy"

_ijerph, 2023, doi:10.3390/ijerph20186739_

Round 1

Reviewer 1 Report (Previous Reviewer 1)

The qualitative data provided in the manuscript seems beneficial to those in the field of psychotherapy. From a methodological approach, I believe that further refinement of Sections 1 and 2 would help to clarify the rationale for and means of gathering the data. But, not being directly involved in the field of psychotherapy myself, I defer to other editors regarding this issue.

The descriptions provided, particularly in Sections 1 and 2, are much improved. However, the authors should proofread the manuscript once more as some grammar and sentence structure still exist.

Author Response

Dear Reviewer, thank you for the valuable suggestions, "The qualitative data provided in the manuscript seems beneficial to those in the field of psychotherapy. From a methodological approach, I believe that further refinement of Sections 1 and 2 would help to clarify the rationale for and means of gathering the data. But, not being directly involved in the field of psychotherapy myself, I defer to other editors regarding this issue." -->

> Thank you for the comments and suggestions, section 1+2 is refined in more detail now, also the sampling and recruitment of expert interviews.

Comments on the Quality of English Language

The descriptions provided, particularly in Sections 1 and 2, are much improved. However, the authors should proofread the manuscript once more as some grammar and sentence structure still exist.

--> A native speaker also included in the first draft checked again. The corrections can be seen in tracked changes in the attachment.

Reviewer 2 Report (Previous Reviewer 2)

I think your manuscript is much better now, congrats!

Just some comments or suggestions:

Introduction: I would introduce some info about psychotherapy effectiveness (in other countries if in Austria is nothing to measure it). And, since in the results appears this idea of "psychotherapists are not effective many times, they don't want difficult patients" and so on, I would state something about the requirements to do psychotherapy in Austria (could be social worker, psychologist...???). 

Line 448: check the sentence, I am not sure it is correct written!

Line 459: theme?

Figure 1: guided expert interviews (a space is needed there)

2.2. Survey method. I suggest stating that all participants signed or given by voice their consent. And maybe (not sure) the problems you had in recruiting should be in Methods too?? (since it is important for understanding the results). 

Lines 181 until 192: you should rephrase this, it's not clear

Line 237: differences.. 

Lines 244-245 rephrase, not clear

Line 266: please check the "" (sometimes one is down, but not consistently)

Line 318: you should be consistent in using (or not) the S.3 (in the first part is not specified). If there are 2 perspectives from just 2 people, maybe would be better to link both results?? If not, then I would suggest stating "this is about participant XXX", and eliminating all the "Interview X" (since there are only 2 people there). I am wondering if the other respondents had ANY sentence or reflection that could fit in these 2 perspectives...

I think the idea in lines 329 and 334 is so important that I suggest amplifying this idea with some info in discussion (free access versus filters)

Lines 356 to 361: a strong idea too, I would amplify it 

Lines 383 to 388: rephrase, good content but not clear

Needs editing!!

Author Response

Dear Reviewer, many thanks for your concise suggestions, we included them all and respond to the text directly point by point:

"Just some comments or suggestions:

Introduction: I would introduce some info about psychotherapy effectiveness (in other countries if in Austria is nothing to measure it). And, since in the results appears this idea of "psychotherapists are not effective many times, they don't want difficult patients" and so on, I would state something about the requirements to do psychotherapy in Austria (could be social worker, psychologist...???). "

--> Thank you for this suggestion, it is added.

Line 448: check the sentence, I am not sure it is correct written!

-->“was and” deleted

Line 459: theme?

 --> “thematization” replaced by theme

Figure 1: guided expert interviews (a space is needed there)

2.2. Survey method. I suggest stating that all participants signed or given by voice their consent. And maybe (not sure) the problems you had in recruiting should be in Methods too?? (since it is important for understanding the results). 

-->This is added.

Lines 181 until 192: you should rephrase this, it's not clear

-->A sentence is added.

Line 237: differences.. 

-->deleted

Lines 244-245 rephrase, not clear

-->Changed.

Line 266: please check the "" (sometimes one is down, but not consistently)

-->quotation marks have been adjusted

Line 318: you should be consistent in using (or not) the S.3 (in the first part is not specified). If there are 2 perspectives from just 2 people, maybe would be better to link both results?? If not, then I would suggest stating "this is about participant XXX", and eliminating all the "Interview X" (since there are only 2 people there). I am wondering if the other respondents had ANY sentence or reflection that could fit in these 2 perspectives...

-->“Interview X” is removed after each quote and the study process is described in more detail.

I think the idea in lines 329 and 334 is so important that I suggest amplifying this idea with some info in discussion (free access versus filters)

-->This is added also in the discussion now.

Lines 356 to 361: a strong idea too, I would amplify it 

--> Done.

Lines 383 to 388: rephrase, good content but not clear

-->Rephrased and explained.

Comments on the Quality of English Language

Needs editing!! --> Done

Please find attached the manuscript in tracked changes and in clean version, kind regards, Henriette Löffler-Stastka

Reviewer 3 Report (Previous Reviewer 3)

The revised version of the manuscript has been significantly improved. 

What is still not clear to this reviewer is the sampling strategy adopted.

Furthermore, I strongly recommend including a description of the core group and Focus group. 

English is clear but some sentences are too long. I would recommend the authors to shorten them. This will certainly make the reading more fluent.

Author Response

Dear Reviewer, thank you for your suggestions, we implemented all of them: 

The revised version of the manuscript has been significantly improved. 

What is still not clear to this reviewer is the sampling strategy adopted.

-->The sampling strategy is described in more detail now.

Furthermore, I strongly recommend including a description of the core group and Focus group. 

-->Both groups are described now.

Comments on the Quality of English Language

English is clear but some sentences are too long. I would recommend the authors to shorten them. This will certainly make the reading more fluent.

--> Sentences are shortened and different ideas are broken up into two sentences.

Round 2

Reviewer 3 Report (Previous Reviewer 3)

The authors deeply reviewed their manuscript and now it can be accepted for publication. 

The authors edited the manuscript, which now reads much more fluently. 

This manuscript is a resubmission of an earlier submission. The following is a list of the peer review reports and author responses from that submission.

Round 1

Reviewer 1 Report

This version of the manuscript is a drastic improvement over the first version. The replacement of tables with text plays a large role in this improvement, but the authors' attention to other details contribute as well.

Although I still question some aspects of the methodologies used, I can attribute these hesitancies to my own preferences for research. Therefore, if other reviewers believe that the manuscript meets the criteria for publication, I will concur. 

I do need to note, however, that I noticed a spelling/grammar mistake toward the end of the manuscript. In line 479, the word "normativ" should be "normative."

none

Author Response

18th July 2023

Dear Reviewer,

Thank you for considering our manuscript. We appreciate your feedback and suggestions, which have helped us improve it. We have implemented your ideas and comments to enhance the quality and clarity of our research. Specifically, we have made revisions based on your suggestions regarding a clearer rationale for study design, research procedure, participant selection, and addressing language concerns. We believe these changes have strengthened the overall coherence.

We genuinely appreciate the time and expertise you have dedicated to reviewing our manuscript. We remain open to any further comments or suggestions you may have.

Thank you once again for your contribution to our study.

Henriette Löffler-Stastka

Reviewer 2 Report

Congratulations for your manuscript

Just some points I would suggest prior to its publication:

- Extensive editing is needed! (see lines 12, 38-40, 42, 47...)

- Line 61 (clients and subjectivity). Needs to be rephrased to be fully understood. And I think it is an important content, so I would check it. 

- Line 97 and 98. This could be used to clarify the objectives of this study. I really liked the way it is expressed.

- Line 158- rephrase this paragraph (sintaxis)

- Line 163. Interview protocol--- I encourage the authors to share their protocol as supplementary material (to be replicated in other countries), or to explain some examples of questions made

- Lines 283 and 284. The cord Codes should be deleted

- DISCUSSION section. Needs to be linked to information about other authors that confirm (or not) your results. One reference is not enough. 

- Lines 422 to 433. No need to explain these other perspectives if they are not explained in the results section. I suggest to remove this information

- Lines 456 to 463. I suggest to check if this paragraph could fit in the last section, to finish with "the future", instead of finishing with the focus on the limitations to the study. 

Extensive editing is needed 

Author Response

(The authors gave the same response as above.)

Reviewer 3 Report

“Expert Perspectives on the Effectiveness of Psychotherapy”

The study conducted by Löffler-Stastka et al. sought to gather insights from 12 experts across various psychotherapy-related fields regarding the effectiveness of psychotherapy in Austria. While the topic is intriguing, I would not recommend publishing it in the International Journal of Environmental Research and Public Health. I have outlined below my considerations:

(1) The abstract fails to provide a concise and accessible summary of the study. It is overly repetitive, lacks specific details about the results obtained, and does not adequately capture the essence of the research.

 (2) The introduction is too general and fails to establish a clear argument. It does not provide a comprehensive overview of recent research on the topic, identify gaps in understanding, or highlight conflicting knowledge. For instance, in Line 21, the authors mention shortages in psychotherapeutic treatments in Austria but do not provide relevant references or up-to-date information.

 (3) Overall, the manuscript is confusing. To improve clarity, I suggest including a flowchart or diagram illustrating the problem, study design, hypotheses, expert recruitment process, interview results, and potential future perspectives and solutions. Additionally, the introduction should clearly specify the targets of the study.

 (4) The authors have not provided any information about the age, gender, ages of professional experience, or nationality of the participating experts, which is an important detail to include.

(5) It would be beneficial to summarize all the interview questions and answers in a table.

(6) It is unclear whether the experiments were conducted blindly. This information should be explicitly addressed.

(7) The discussion section merely summarizes the findings and lacks adequate references to previous studies conducted in other countries. Furthermore, the experts' answers do not seem to be used to propose any potential solutions.

(8) Although the Limits section acknowledges the study's main limitation, namely the small number of participants, I believe it is crucial to recruit additional experts to enhance the study's power and significance.

Based on these concerns, I strongly urge the authors to revise the manuscript, increase the sample size, provide a clearer rationale for the study, and incorporate a flowchart illustrating the study design.

I am very sorry, but I think that this study cannot be published in the present form.

English must be improved

Author Response

(The authors gave the same response as above.)
